# Metrics of high cofluctuation and entropy to describe control of cardiac function in the stellate ganglion

**Nil Z Gurel**[1]*[†], **Koustubh B Sudarshan**[2][†], **Joseph Hadaya**[1,3], **Alex Karavos**[2], **Taro Temma**[1], **Yuichi Hori**[1], **J Andrew Armour**[1], **Guy Kember**[2], **Olujimi A Ajijola**[1,3]

[1]UCLA Cardiac Arrhythmia Center and UCLA Neurocardiology Research Program of Excellence, Los Angeles, United States; [2]Department of Engineering Mathematics and Internetworking, Dalhousie University, Nova Scotia, Canada; [3]UCLA Molecular, Cellular, and Integrative Physiology Program, Los Angeles, United States

**Abstract** Stellate ganglia within the intrathoracic cardiac control system receive and integrate central, peripheral, and cardiopulmonary information to produce postganglionic cardiac sympathetic inputs. Pathological anatomical and structural remodeling occurs within the neurons of the stellate ganglion (SG) in the setting of heart failure (HF). A large proportion of SG neurons function as interneurons whose networking capabilities are largely unknown. Current therapies are limited to targeting sympathetic activity at the cardiac level or surgical interventions such as stellectomy, to treat HF. Future therapies that target the SG will require understanding of their networking capabilities to modify any pathological remodeling. We observe SG networking by examining cofluctuation and specificity of SG networked activity to cardiac cycle phases. We investigate network processing of cardiopulmonary transduction by SG neuronal populations in porcine with chronic pacing-induced HF and control subjects during extended in-vivo extracellular microelectrode recordings. We find that information processing and cardiac control in chronic HF by the SG, relative to controls, exhibits: (i) more frequent, short-lived, high magnitude cofluctuations, (ii) greater variation in neural specificity to cardiac cycles, and (iii) neural network activity and cardiac control linkage that depends on disease state and cofluctuation magnitude.

**\*For correspondence:**
gurelnil@gmail.com

[†]These authors contributed equally to this work

**Competing interest:** The authors declare that no competing interests exist.

## Editor's evaluation

The paper thoroughly examines the role of the stellate ganglia in the control of cardiac rhythmicity. The empiric findings on stellate-ganglia-mediated regulation of cardiac activity in the presence of chronic cardiac failure, in particular, are translationally relevant and could be the basis of future drug-based interventions.

## Introduction

Neural control of cardiac function involves adaptive adjustment of mechanical and electrical activity to meet the organism's demand for blood flow. This cardioneural control scheme consists of neural populations in the central, peripheral, and intrinsic cardiac nervous systems. Interactions among components of the cardiac nervous system highlight that these neural populations work in concert, rather than as independent, singular processing units (*Ardell et al., 2016*). From an information processing standpoint, the operation of these interconnected neural networks has evolved to coordinate cardiac function on a beat-by-beat basis, producing the 'functional' outputs of this control scheme such as blood pressure, heart rate, or respiratory pressure (RP) and rate. Localized adaptations in the

cardioneural network in response to pathology can cause an evolution of global network properties with heightened risk of poor outcomes without measurable evidence from these functional outputs (*Deyell et al., 2015*; *Kember et al., 2013*).

There is a current focus on understanding cardioneural network processing within the stellate ganglion (SG), a collection of nerves serving as the major source of sympathetic input to the heart (*Mehra et al., 2022*). The SG (located in either side of the neck) operates as an integrative layer within the control hierarchy where it processes central cardiac inputs to the heart, receives cardiac feedback, and projects efferent control outputs to the heart. In pathological states such as heart failure (HF), morphological and neurochemical remodeling of SG neurons have been reported in both animal models (*Ajijola et al., 2013*; *Han et al., 2012*; *Ajijola et al., 2015*; *Nakamura et al., 2016*) and in humans (*Ajijola et al., 2020*; *Ajijola et al., 2012b*). Due to its key role in proarrhythmic neural signaling and convenience in surgical accessibility, clinical interventions targeting SG are used to treat various cardiovascular conditions (*Vaseghi et al., 2012*; *Vaseghi et al., 2017*; *Ajijola et al., 2012a*). It has also been established that an enhanced cardiac sympathetic afferent reflex contributes to sympathoexcitation and pathogenesis of HF (*Wang and Zucker, 1996*; *Ma et al., 1997*; *Chen et al., 2015*; *Wang et al., 2017*; *Wang et al., 2008*; *Wang et al., 2014*; *Gao et al., 2005*; *Gao et al., 2007*). Despite these novel interventions and general understanding, SG clinical therapy will remain largely unexplored without greatly improved understanding of SG neuronal information processing in healthy versus pathological states. Prior studies examining the SG neural activity have been limited to in vivo extracellular recordings (*Armour, 1983*; *Armour, 1986*; *Armour et al., 1998*; *Yoshie et al., 2020*; *Yoshie et al., 2018*).

Recently, we explored network processing of cardiopulmonary transduction by SG neuronal populations in healthy porcine, defining a novel metric 'neural specificity' that measures specificity of neural firing patterns to cardiopulmonary signals (*Sudarshan et al., 2021*). This metric is contrastive and a measure of the difference between the probability density function (PDF) of neural 'sampling' of a control target relative to the same in the random sampling limit. While the target, left ventricular

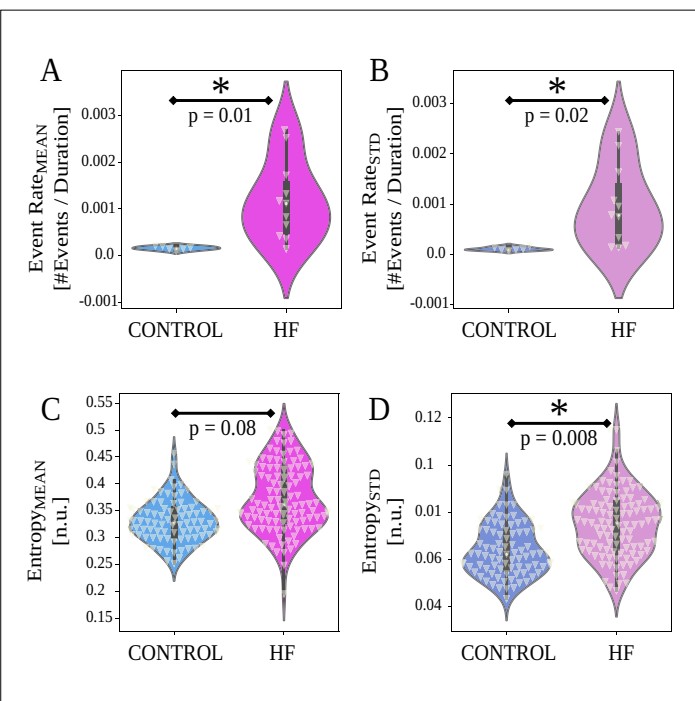

**Figure 1.** Event rate (*ER*) and entropy results between control and heart failure (HF) animals (horizontal axes). White triangles indicate data points. (**A, B**) HF group animals show higher $ER_{MEAN}$ and $ER_{STD}$ compared to control group ($p < 0.05$). (**C, D**) HF group animals show higher entropy variability (*Entropy_STD*, **Equation 2**, $p = 0.008$, in (**D**)), and no difference in *Entropy_MEAN* (**Equation 2**, $p = 0.08$, in (**C**)). For *ER*, $p$ values are from two-sample $t$-test or Wilcoxon rank-sum tests, depending on normality. For entropy, $p$-values are from linear mixed effects (LMEs, **Equation 2**) detailed in Materials and methods.

pressure (LVP) considered here is periodic this is not a necessary condition for use of the specificity metric; it is also applicable to aperiodic signals in an event-based fashion.

In the current work, we investigate differences in information transfer between control and HF porcine models with multi-channel electrode arrays. We first uncover network-level spatiotemporal dynamic signatures by quantifying short-lived high cofluctuation events in neural activity. Second, we study coherence and consistency in the evolution of neural specificity with respect to the control target. Third, we expose differences in neural specificity and its coherence and consistency, via entropy, inside and outside cofluctuation events. These differences are considered for control and HF models and quantify differences in the maintenance of function between these groups.

## Results

Neural activity was measured over 16 channels along with simultaneous LVP for approximately 6 hr of continuous recordings per animal. Representative neural activity recording for a single channel, LVP, and representative spike trains are displayed for control and HF animals in Figure 6A. A total of 17 Yorkshires (6 control, 11 HF; Figure 6D) underwent the terminal experiment described in Figure 6E. Upon the signal processing pipeline described above, we computed two event rate measures per animal as the final product representing the cofluctuations ($ER_{MEAN}$, $ER_{STD}$). As the metric representing the neural specificity, we computed two entropy measures per channel ($Entropy_{MEAN}$, $Entropy_{STD}$), resulting in a total of 16 $Entropy_{MEAN}$ and 16 $Entropy_{STD}$ per animal. Finally, we used these metrics to quantify: (i) neural population dynamics (i.e., $ER_{MEAN}$, $ER_{STD}$), (ii) neural specificity to target LVP, or cardiac control (i.e., $Entropy_{MEAN}$, $Entropy_{STD}$), and (iii) linkage between neural population dynamics and specificity (i.e., $Entropy_{MEAN,EVENT}$, $Entropy_{STD,EVENT}$).

### SG in HF exhibits high event rate

*Figure 1A-B* shows event rate outcomes grouped by HF models and controls. HF animals show significantly higher event rates compared to control animals for both $ER_{MEAN}$ ($p = 0.011$, effect size

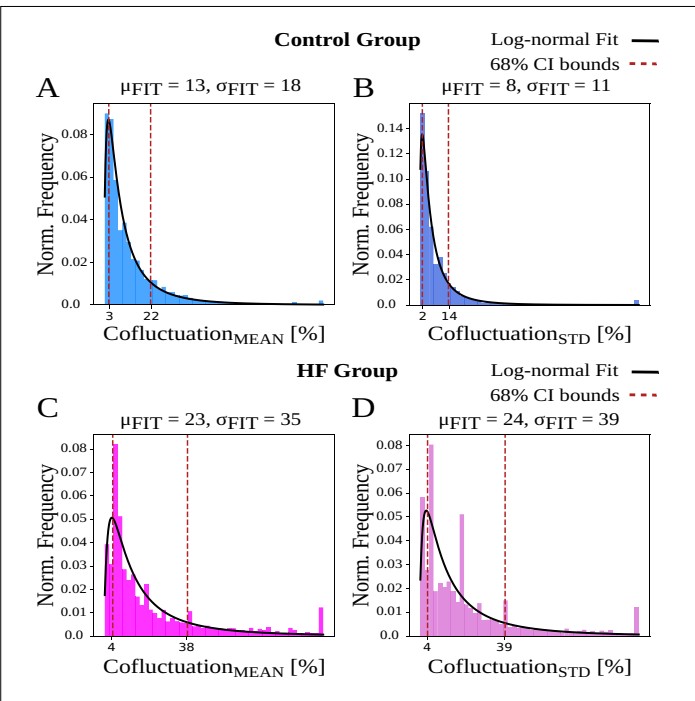

**Figure 2.** Cofluctuation histograms (calculated from mean or standard deviation of sliding spike rate, referred as *Cofluctuation_{MEAN}* and *Cofluctuation_{STD}*, respectively) and log-normal fits for each animal group. $\mu_{FIT}$ and $\sigma_{FIT}$ are the respective mean and standard deviation (STD) of fitted distribution, used for 68% confidence interval bounds. (**A, B**) Control animals have narrower bounds and represent a better fit to log-normal distribution. (**C, D**) Heart failure (HF) animals display more heavily skewed distributions that indicate heavy tails.

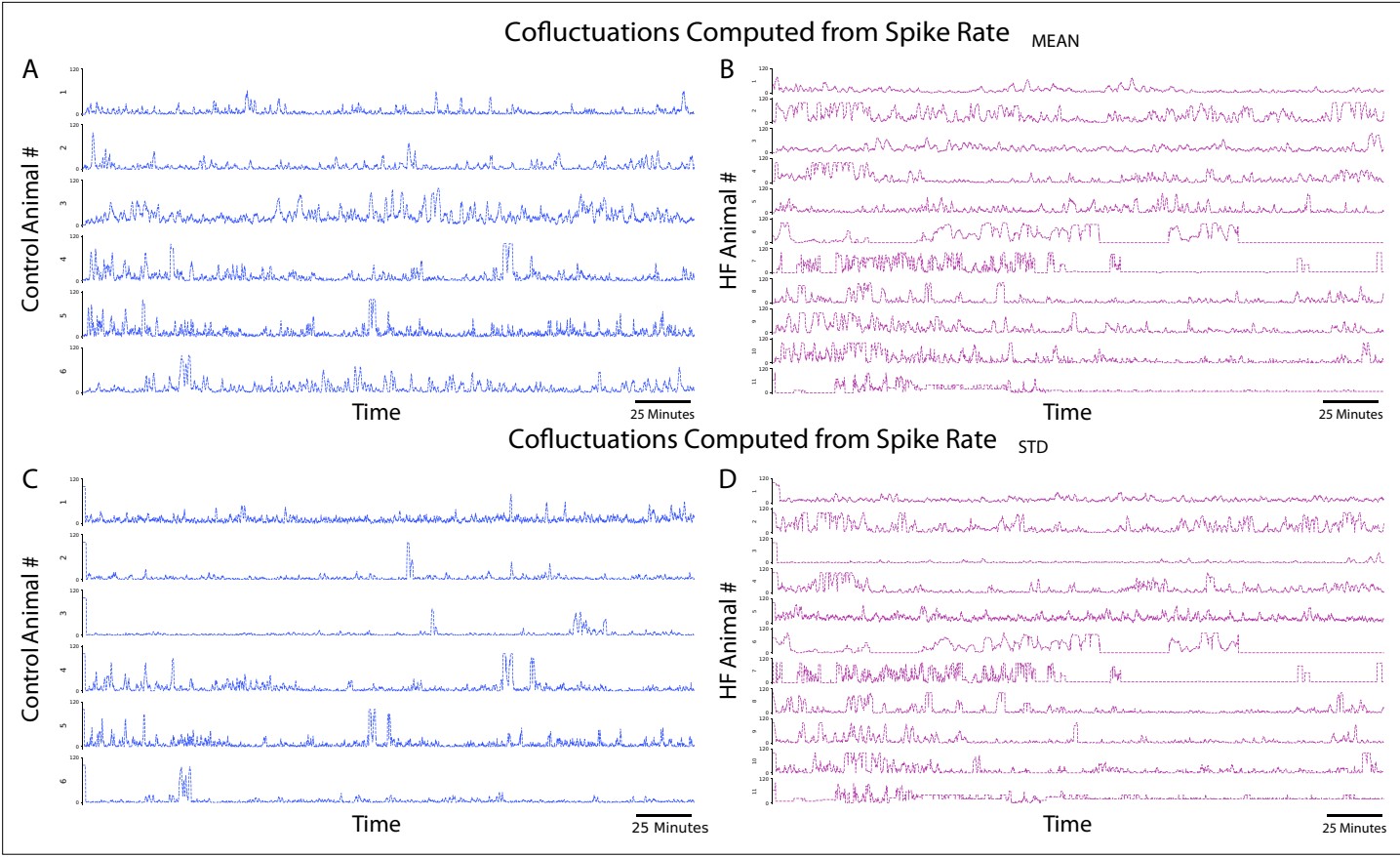

**Figure 3.** Cofluctuations time series at convergent $C$ values for each animal. (**A**) Cofluctuations from coactivity calculation from mean of sliding spike rate for control animals. (**B**) Cofluctuations from coactivity calculation from mean of sliding spike rate for heart failure (HF) animals. (**C**) Cofluctuations from coactivity calculation from standard deviation of sliding spike rate for control animals. (**D**) Cofluctuations from coactivity calculation from standard deviation of sliding spike rate for HF animals.

$d = 1.59$, $ER_{MEAN,HF} = 0.0012 evts/s$, $ER_{MEAN,Controls} = 0.0002 evts/s$) and $ER_{STD}$ ($p = 0.023$, $d = 1.48$, $ER_{STD,HF} = 0.001 evts/s$, $ER_{STD,Controls} = 0.0001 evts/sec$). The cofluctuation time series for each animal is depicted in Figure 3, where the event time series are computed. The 'events' or short-lived intervals where high cofluctuations exist are shown as level 1, leading to the event time series in Figure 4. We observe that the cofluctuations are more localized in HF animals with greater heterogeneity.

## HF animal models have heavy-tailed cofluctuation distributions

We qualitatively explored the statistical distribution of the cofluctuation time series. *Figure 2* shows log-normal fits for each animal group for $Cofluctuation_{MEAN}$ and $Cofluctuation_{STD}$ time series, along with 68% confidence interval (CI) bounds, mean of fit ($\mu_{FIT}$) and standard deviation of fit ($\sigma_{FIT}$). Control animals (*Figure 2A–B*) exhibit narrow CIs, lower ($\mu_{FIT}$) and ($\sigma_{FIT}$) values, and tighter log-normal fits. In contrast, HF animals (*Figure 2C–D*, *Figure 3*, *Figure 4*) exhibit wider CIs, higher ($\mu_{FIT}$) and ($\sigma_{FIT}$) values, and poorer log-normal fits. Of note, HF animals have heavy tails ranging further outside of confidence bounds.

## SG shows greater variation in neural specificity to LVP in HF

We next examined the neural specificity to LVP, quantified by entropy measures in *Equation 2*. *Figure 1C–D* shows $Entropy_{MEAN}$ and $Entropy_{STD}$, grouped by animals. Compared to the control group, SG of HF animals exhibited significantly higher $Entropy_{STD}$ (variation in entropy, *Figure 1D*, adjusted $\beta = 0.01$ n.u., 95% $CI = \pm 0.01$ n.u., $d_{RM} = 0.73$, $p = 0.009$). However, there is no significant difference in $Entropy_{MEAN}$ (mean entropy) between animal groups. (*Figure 1C*, $\beta = 0.04$ n.u., $\pm 0.05$ n.u., $d_{RM} = 0.82$, $p = 0.087$).

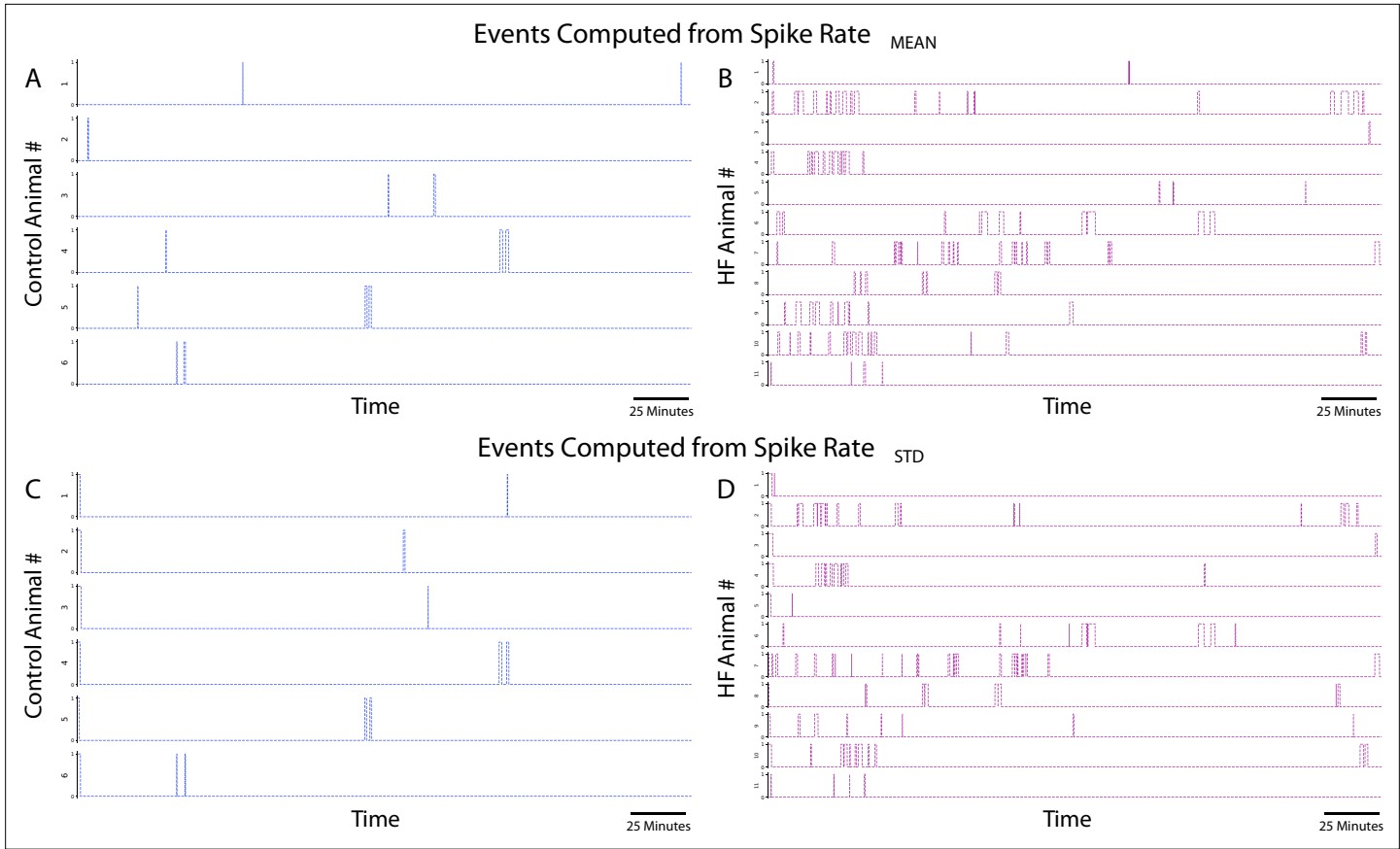

**Figure 4.** Events time series at convergent $(C, T)$ pairs for each animal. (**A**) Events from coactivity calculation from mean of sliding spike rate for control animals. (**B**) Events from coactivity calculation from mean of sliding spike rate for heart failure (HF) animals. (**C**) Events from coactivity calculation from standard deviation of sliding spike rate for control animals. (**D**) Events from coactivity calculation from standard deviation of sliding spike rate for HF animals.

## Neural network activity and cardiac control linkage depends on animal group and cofluctuation magnitude

We explored the nature of cardiac control inside and outside short-duration regions of high cofluctuation, that is, 'events', characterized by strongly coherent stellate neural activity patterns. Insight into how these events may be relevant to cardiac control is considered here in the context of how control differs inside and outside events and termed 'event entropy'.

First, we studied the extent to which event entropy differs inside and outside of events (*Figure 5A, C*, event type as fixed effect in Eq. (2)). Second, we studied whether event entropy is sensitive to the animal type characterized here as control or HF (*Figure 5B, D*, animal type as fixed effect in Eq. (9)).

Regardless of the animal group, $Entropy_{MEAN,NON-EVENT}$ significantly exceeds $Entropy_{MEAN,EVENT}$ (*Figure 5A*, $\beta = 0.007$ n.u., $\pm0.004$ n.u., $d_{RM} = 0.07$, $p < 0.001$). Similarly, $Entropy_{STD,NON-EVENT}$ significantly exceeds $Entropy_{STD,EVENT}$ (*Figure 5C*, $\beta = 0.01$ n.u., $\pm0.002$ n.u., $d_{RM} = 0.29$, $p < 0.001$). An examination of the contribution of each animal group showed no significant difference between groups for $Entropy_{MEAN,EVENT}$ (*Figure 5B*, $\beta = 0.06$ n.u., $\pm0.0$ n.u., $d_{RM} = 1.13$, $p = 0.07$). On the other hand, HF animals exhibited an increase in $Entropy_{STD,EVENT}$ compared to control animals (*Figure 5D*, $\beta = 0.02$ n.u., $\pm0.02$ n.u., $d_{RM} = 0.75$, $p = 0.012$). These analyses imply that the linkage between neural network function and cardiac control differ inside and outside of cofluctuation events and between animal groups in the SG.

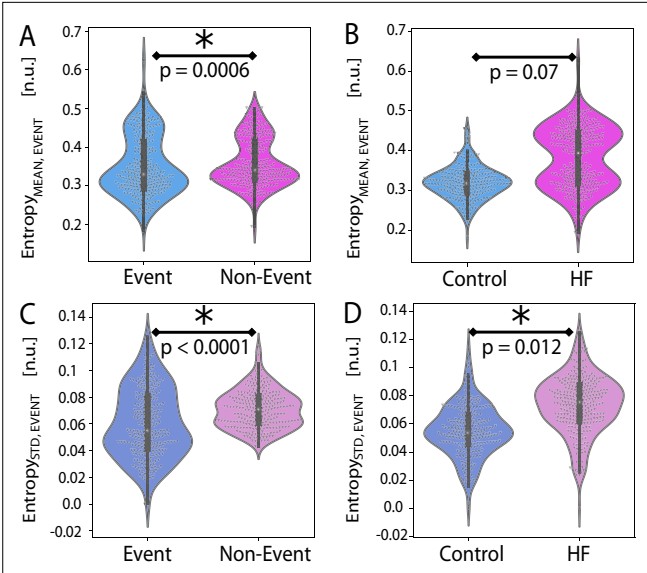

**Figure 5.** Event entropy *Equation 2* investigation involved consideration of entropy values inside and outside of event regions. (**A**) There is significant difference between in $Entropy_{MEAN,EVENT}$ and $Entropy_{STD,EVENT}$ across all animals ($p = 0.0006$). (**B**) There is no significant difference in $Entropy_{MEAN,EVENT}$ between animal groups ($p = 0.07$). (**C**) There is significant difference in $Entropy_{STD,EVENT}$ between events and non-events across all animals ($p < 0.0001$). (**D**) There is significant difference in $Entropy_{STD,EVENT}$ between animal groups ($p < 0.012$).

## Discussion

In this work, we performed a novel investigation of SG neural population dynamics and neural specificity to continuous LVP in control and HF Yorkshire pigs. The methods in this work are intended to measure the way population neural activity relates to closed-loop control of a target and how that computation changes in diseased states. This was applied here to closed-loop control of cardiac output where the assumed target was LVP.

The methods in this work involved.

- *Neural specificity*: A measure of bias in neural activity toward 'sampling' of specific target states. The target specificity is a contrastive measure that compares neural sampling of a target relative to random sampling of the same target.
- *Neural specificity coherence*: Entropy of neural specificity was used to measure coherence of neural specificity as a function of time.
- *Cofluctuation events*: The degree of coactivity in the dynamics of the mean and its standard deviation was measured between pairs of channels from minimum to maximum physical separation and this exposed short-duration 'events' when cofluctuation was unusually high.
- *Event entropy*: Functional significance of cofluctuation events was evaluated by comparing differences in the degree of neural specificity coherence inside and outside of events.

### Prevalence of short-lived cofluctuations in SG activity in HF

In prior work, we identified neural specificity toward near-peak systole of the LVP waveform in control animals (*Sudarshan et al., 2021*). Application of this metric and the construction of a related coherence measure provided insight into differences in neural processing dynamics between control and HF animals. Our results show that cardiac control exerted within diseased states has greater variation in entropy and thus less consistency for HF animals compared to control animals. This finding may extend to other pathologies for which the cardiac control hierarchy is disrupted.

### Neural network activity is linked to cardiac control

Based on the effect size ($d_{RM}$), event entropy magnitude appears to be higher with greater variation observed in HF animals compared to control animals (*Figure 5B-D*). This implies a level of increased unpredictability and increased difficulty in cardiac control for animals in HF over control animals.

A limitation of this result is that the effect sizes for event versus non-event comparisons are small to medium, which potentially indicates a larger study is necessary to better understand the physiological contributions from event type. Another limitation of the study lies in the absence of multiple-class pathologies (i.e., different HF models or other reproducible models) and in the absence of stratified pathologies (i.e., animal models with varying degrees of HF). Measurement of these neurocardiac metrics during slow, quasi-static application of clinically relevant stressors (*Akeju and Brown, 2017*; *Chamadia et al., 2019*) should provide unique opportunities to investigate unresolved questions. Future studies should focus on expanding the data set to examine how these metrics change with varying pathologies or varying disease models. We also cannot exclude possible effects of general anesthesia, open chest, and open pericardial effects on our findings, though the effects are likely consistent across the groups studied in the same manner.

## Conclusion

In this study, we looked, for the first time to our knowledge, at long-term studies of in vivo cardiac control in baseline states. The baseline states provide unique signatures that differentiate animals with HF and controls. We discovered the inputs (i.e., neural signals) and outputs (i.e., blood pressure) are linked, which led us to develop metrics to analyze the dynamical state of this networked control (*Gurel et al., 2022*). The primary observation has been that event-based processing within the SG and its relationship to cardiac control is strongly modified by HF pathology. Our analysis is pointing to HF being best considered as a spectrum rather than a binary state. The magnitude of cofluctuation and neural specificity may give us a measure of the degree of HF and insight into the extent to which cardiac control is compromised with respect to neural specificity and/or cofluctuation. Future therapies may benefit from being able to infer the degree of HF in terms of neural markers as represented in this work, in a less invasive way. Intriguing connections involve the alignment of our work with a growing consensus in neuroscience. Spatiotemporal changes in neural activity and linkages with control targets are associated with behavioral changes and the onset and development of specific pathologies. For instance, spatiotemporal brain-wide cofluctuations were reported to reveal major depression vulnerability (*Hultman et al., 2018*). Neural ensembles were linked to visual stimuli in mice (*Miller et al., 2014*). Another study reported that brain's functional connectivity is driven by high-amplitude cofluctuations and that these cofluctuations encode subject-specific information during experimental tasks (*Zamani Esfahlani et al., 2020*). Similar cofluctuations were also reported to inform olivary network dynamics in the form of state changes in learning new motor patterns in mice (*Wagner et al., 2021*). Unique co-activation patterns in spontaneous brain activity indicated a signature for conscious states in mice (*Gutierrez-Barragan et al., 2022*). Global brain activity has also been linked to higher-level social behaviors (*Mague et al., 2022*). These parallel conclusions in cardiac and neuroscience studies indicate similar experimental methods used to measure neural integration relative to control targets. Such measurements may be instrumental to design and assess the efficacy of neurally based clinical interventions both at the level of the brain and the SG.

## Materials and methods
### Animal experiments

*Figure 6* presents the conceptual overview and study design. The study was performed under a protocol approved by the University of California Los Angeles (UCLA) Animal Research Committee (ARC), in compliance with the UCLA Institutional Animal Care and Use Committee (IACUC) guidelines and the National Institutes of Health (NIH) Guide for the Care and Use of Laboratory Animals (Protocol: ARC 2015-022). *Figure 6D–E* summarizes the studied animal groups and experimental pipeline. Male Yorkshire pigs ($n = 17$) weighing $57.5 \pm 12 kg$ were studied as control ($n = 6$) and HF model ($n = 11$) groups. For SG neural data collection, the animals were sedated with tiletamine and zolazepam (Telazol, 4–8 mg/kg) intramuscularly, intubated, and maintained under general anesthesia with inhaled isoflurane (2%). Continuous intravenous saline ($8 - 10 ml/kg/h$) was infused throughout the protocol and animals were temperature maintained using heated water blankets ($37 - 38^o C$).

Median sternotomy by an incision down the midline of the entire sternum was performed to have a wide view of the thoracic region (*Figure 6A*). The pericardium was opened to expose the heart and both stellate ganglia. After surgical procedures, animals were transitioned to alpha-chloralose

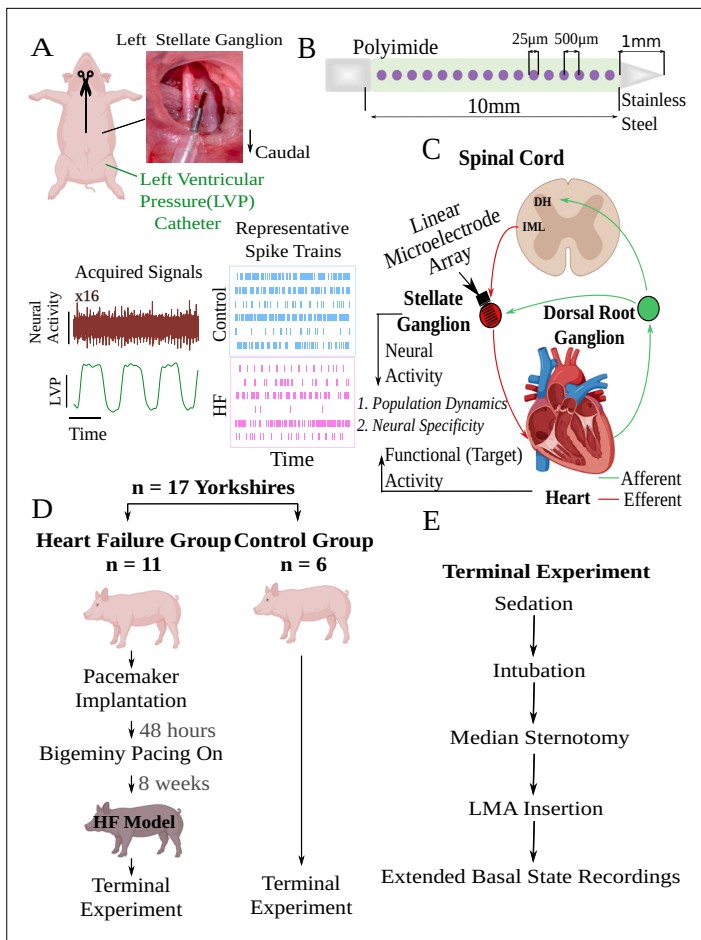

**Figure 6.** Experimental workflow and overall concept. (**A**) A linear microelectrode array (LMA) was inserted to the left stellate ganglion (SG) for each animal. A total of 16 channels of neural activity were collected along with simultaneous left ventricular pressure (LVP). Representative spike trains are displayed for each animal group. (**B**) Specifications of the LMA. (**C**) Conceptual representation of this work. SG receives efferent and afferent information from the spinal cord's intermediolateral complex (IML) and dorsal root ganglion, respectively, and transmits efferent information to the heart. In this work, we investigate neural activity and its relationship to cardiac function as represented by a control target such as LVP. (**D**) Among 17 Yorkshire pigs, 11 had heart failure induced by ventricular pacing, and 6 were in control group. SG recordings were collected at terminal experiments for both groups. (**E**) Experimental flow describing the surgical preparation for the recordings. DH, dorsal horn.

anesthesia ($6.25mg/125ml$ for bolus, $20 - 35ml/kg$ or titrated to effect for maintenance) with supplemental oxygen ($2L/min$) for in vivo neural recordings from the left SG. The left carotid artery was exposed, and a pressure catheter (SPR350, Millar Inc, Houston, TX) was inserted to continuously monitor LVP. Additionally, three-lead surface electrocardiogram (ECG) and RP were monitored continuously, and sampled at $1kHz$. Arterial blood gas contents were monitored at least hourly to ensure appropriate experimental conditions. At the end of the protocol, animals were euthanized under deep sedation of isoflurane and cardiac fibrillation was induced.

The HF model was created with implanted pacemakers (Viva Cardiac Resynchronization Therapy–Pacemaker, Biotronik, Lake Oswego, OR), as previously described (*Hori et al., 2021*), and summarized in *Figure 6D*. After implantation, animals had a recovery period of 48 hr and chronic bigeminy pacing was initiated from the right ventricle. This process produces premature ventricular contractions (PVCs) which lead to cardiomyopathy, also known as PVC-induced cardiomyopathy (*Sadron Blaye-Felice et al., 2016*). To confirm the progression of cardiomyopathy, echocardiography was performed, before and after implantation. After the animals have been confirmed to have cardiomyopathy (referred as HF animals) at 8 weeks after implantation, surgical procedures described in *Figure 6E* were performed, and extracellular recordings were obtained from the left SG, shown in

*Figure 6A*. It should be noted that a subset of HF animals (*n* = 6) underwent an intervention, epicardial application of resiniferatoxin (RTX) to study its effects on the progression of cardiomyopathy as a separate study. However, no significant effect of RTX was noted in any of the echocardiographic, serum, physiological, and autonomic tests (*Hori et al., 2021*). Hence, in this work, we combined RTX-treated HF animals with untreated HF animals.

We confirmed the RTX depleted the afferents by analyzing both structural and functional data (*Hori et al., 2021*). Structural depletion was proven with immunohistochemistry studies of the left ventricle (LV) and T1 dorsal root ganglion (DRG). Calcitonin gene-related peptide (CGRP)-immunoreactive fibers, a marker of sensory afferent nerves, was significantly reduced within the nerve bundles located in the LV for the RTX-treated group. Furthermore, the depletion of cardiac transient receptor potential vanilloid-1 (TRPV1) afferents was confirmed by the significant reduction of CGRP-expressing neurons in DRG. Functional depletion was proven by the response to the agonist of TRPV1 channel bradykinin and capsaicin. The RTX-treated group had a significantly lower LV pressure (LVP) response in the application of bradykinin and capsaicin, indicating that elimination of cardiac sympathetic afferent reflex was accomplished by RTX application in each case.

## SG neural recordings and experimental protocol

For each animal, a 16-channel, linear, single-shank microelectrode array (LMA, Microprobes, Gaithersburg, MD) was inserted in the craniomedial pole of the left SG (*Figure 6A*). The LMA consisted of a polyimide tube of $0mm$ that contains recording sites, and a stainless steel tip of $1mm$ (*Figure 6B*). Polyimide tube hosted a total of 16 platinum-iridium recording sites with $25\mu m$ radius, separated by $500\mu m$ intra-electrode spacing. A microelectrode amplifier (Model 3600, A-M Systems, Carlsborg, WA) was used to amplify (gain of $1000 - 2500$) and filter ($300Hz - 3kHz$ band-pass filter) the acquired

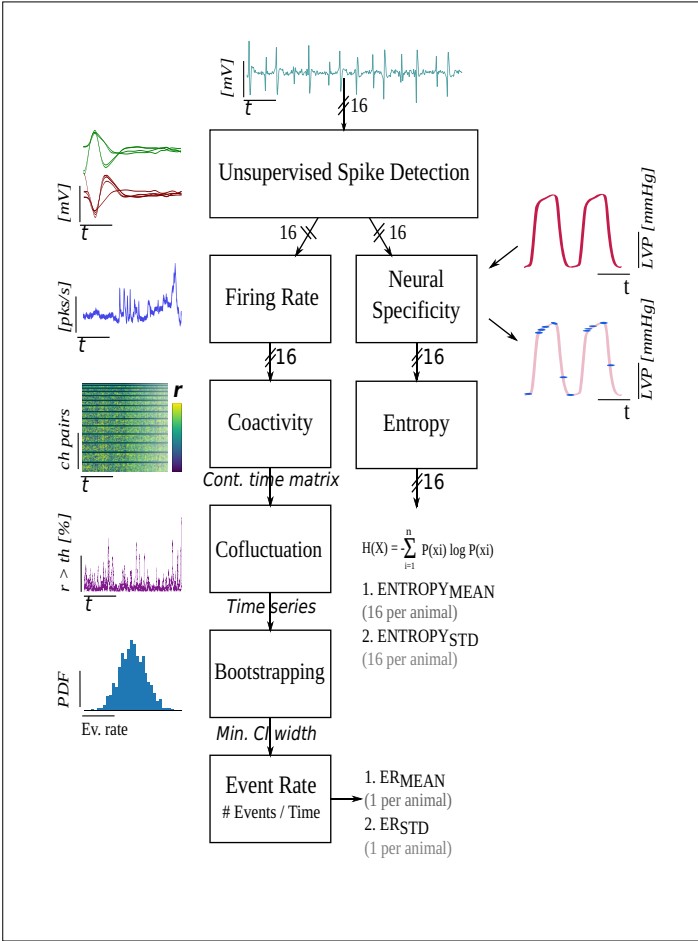

**Figure 7.** Signal processing block diagram.

signals. The signals were transferred to a data acquisition platform (Power 1401, Cambridge Electronic Design, Cambridge, UK) and recorded using Spike2 software (Cambridge Electronic Design, Cambridge, UK). All data were processed in Python and MATLAB. Increases in spike rate occur within 90 min of electrode insertion, hence a stabilization time of approximately 3 hr is required after the insertion takes place (*Sudarshan et al., 2021*).

It should be noted that our study deals with multi-electrode recordings of the closest neural populations to the electrode array. The earliest fundamental studies probing into cardiac nervous system used single-unit recordings, for which the target neurons should be isolated and appropriate low-impedance conductors should be used for obtaining high-quality neural signals. Unlike these early studies, we used multi-unit (16-channel) electrode arrays to monitor the ensemble behaviors of SG neural populations. This experimental shift from single-unit to multi-unit recording has gained interest in the recent years in neurocardiology and neuroscience communities, offering an experimental view to the ensemble behaviors of neural populations (*Gurel et al., 2022*).

## Signal processing and time-series analysis

### Signal processing pipeline

A high-level description of the signal processing pipeline is in *Figure 7*. In summary, Pearson's cross-correlation is used to construct the coactivity matrix as the collection of cross-correlations between all possible channel pairs. The coactivity matrix is computed at each timestamp and associated with a window of past neural activity (*Figure 7*, 'Coactivity' block). This computation yields a causal sliding window of coactivity matrices referred to as the 'coactivity time series'.

Discrete events of high cofluctuation occurring in the coactivity time series are defined using two thresholds: (i) the coactivity time series is mapped to a univariate 'cofluctuation time series' where, at each timestamp, the percentage of coactivity matrix members exceeding a threshold $C$ is found, and (ii) discrete 'events' are defined as those timestamps when up-crossings of the cofluctuation time series through a second threshold $T$ occur. The method used to choose the $(C, T)$ pair, detailed in this section, generates discrete event timestamps and allows for the computation of the event rate ($ER$) mean and standard deviation ($STD$) statistics, which are used later in the statistical analyses. These cofluctuation events are regions that expose shifts in neural processing within the SG. These events are linked to function through the consideration of how neural specificity differs inside and outside cofluctuation events in control and HF animals.

The relationship between a control target such as LVP and neural activity at each channel is quantified via a continuously varying neural specificity (*Sudarshan et al., 2021*; *Figure 7*, 'Neural Specificity' block). The neural specificity is contrastive since it is the difference between the PDF of neural sampling of a target and the same found from random sampling. The neural activity in the SG is known to be a mixture of afferent, efferent, and local circuit activity derived from local circuit neurons with inputs from multiple sources. It in this sense that we define neural computation; when we observe the specificity to the target operating above or below the random sampling limit. Neural specificity is a multivariate signal measured across multiple target states at each channel as a function of time. This is reduced, for each channel, to a univariate time series by constructing its coherence in terms of entropy. The evolution of coherence in time provides access to the dynamics or consistency of neural computation. Detailed information about each signal processing step is provided in this section. Appendix 1 contains material detailing the mathematical aspects of the analysis. As stated in the signal processing block diagram, our outcome measures are event rate, entropy, and event entropy. These metrics are developed in Appendix 1.

### Unsupervised spike detection

We use a competitive, adaptive threshold, unsupervised approach for neural spike detection (*Sudarshan et al., 2021*). The algorithm initializes plus and minus barriers at the plus or minus signal maximum amplitude. The barriers are respectively lowered or raised until the plus or minus barrier 'wins the competition' and is the first to yield a minimal number of crossings. Detected spike regions are masked as a zero signal and the process repeated with barrier sizes further reduced in subsequent iterations. The competition is halted when one barrier is first to reach a minimal barrier height.

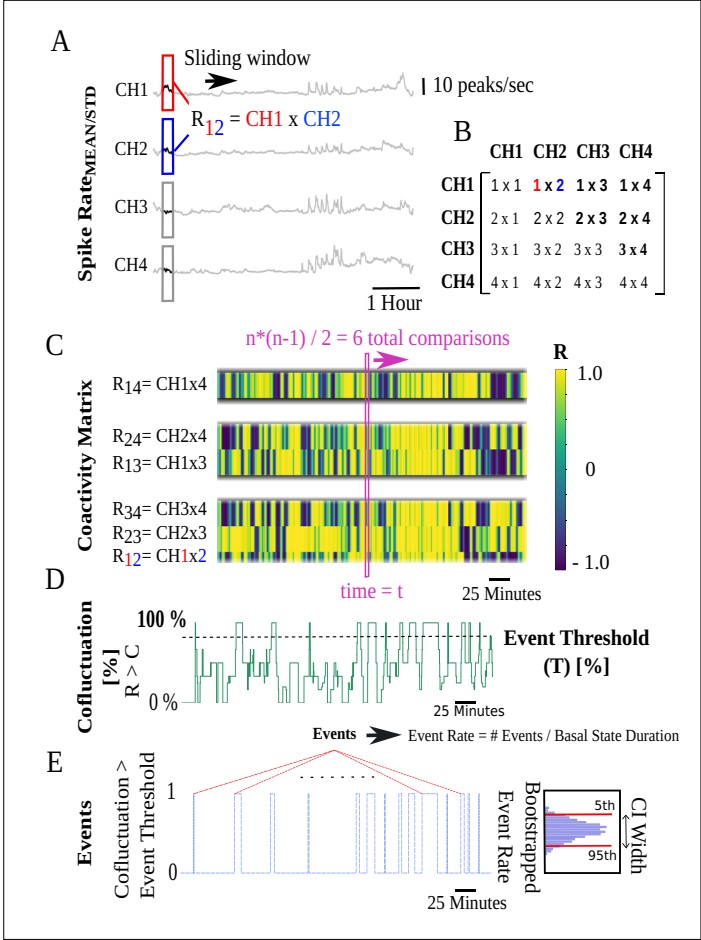

**Figure 8.** Coactivity matrix and event rate ($ER$) computation pipeline illustrated for four channels. (**A**) Pearson's cross-correlation coefficients ($R$) of pairwise sliding spike rate windows for four channels. (**B**) $R$ values for cross-correlation of '$i$' and '$j$' channels are stored in the '$i$' and '$j$' positions of a 4×4 matrix at each timestamp. (**C**) Coactivity matrix super-diagonals at each timestamp are vertically stacked. At each timestamp $t$, $n(n-1)/2 = 6$ unique cross-correlations are possible corresponding to super-diagonals with 3, 2, and 1 members. The y-axis begins at the bottom with the three members of the first super-diagonal followed by subsequent super-diagonals. Colors represent the $R$ value. (**D**) Cofluctuation time series are the percentage of Pearson's $R$ values at each timestamp that exceed a cofluctuation threshold ($C$). (**E**) Discrete events correspond to time intervals when cofluctuations show an up/down crossing through an event threshold ($T$).

## Code availability

Supporting Apache License codes are at GitHub (https://github.com/Koustubh2111/Cofluctuation-and-Entropy-Code-Data; copy archived at swh:1:rev:1ca5e9ce38151715cfa1aeb3d38f3bdbbf796a05; *Koustubh, 2022*).

## Data set and statistical analysis

Statistical analyses are performed in MATLAB Statistics & Machine Learning Toolbox (version R2021a) and Python SciPy Library (version 3.8.5).

## Sample size breakdown

Two channels were excluded from two animals due to insufficient signal quality. Within event rate analyses, all animals had sufficient neural data ($n = 17$ animals, 6 control, 11 HF). Entropy analyses for three HF animals were excluded due to insufficient LVP quality resulting in $n = 14$ animals (6 control, 8 HF).

## Outcome measures

Within the signal processing pipeline described in *Figure 8*, the event rate measures, $ER_{MEAN}$ and $ER_{STD}$, are used to summarize the cofluctuation time series for each animal. A mean and standard deviation of the 16 channel-wise entropy time series results in 32 measures of entropy per animal (16 for $Entropy_{MEAN}$ and 16 for $Entropy_{STD}$ per animal).

## Statistical analysis

For variables that result in a single number per animal (such as $ER_{MEAN}$ and $ER_{STD}$, *Figure 1A–B*), independent samples *t*-tests or Wilcoxon rank-sum tests are respectively used for normal or non-normal data (normality assessed by Shapiro-Wilk) to quantify differences between animal groups.

For variables that have multiple variates per animal (such as $Entropy_{MEAN}$ calculated from multiple channels, *Figure 1C–D*), mixed effects models are constructed in the MATLAB Statistics and Machine Learning Toolbox (*Pinheiro and Bates, 1996*; *MATLAB, 2021*). $Entropy_{MEAN}$ and similarly $Entropy_{STD}$ (not shown) and $Entropy_{MEAN,EVENT}$ and similarly $Entropy_{STD,EVENT}$ (not shown) are modelled via mixed effects as, 1| indicates random effects,

$$Entropy_{MEAN} = Animal\ Type + (1|channel) + (1|animal\ ID) \tag{1}$$

$$Entropy_{MEAN,EVENT} = Event\ Type + Animal\ Type + CoactivityType + (1|channel) + (1|animal\ ID) + (1|Entropy_{MEAN}) \tag{2}$$

In *Equation 2*, and depicted in *Figure 1C–D*, the computed metric $Entropy_{MEAN}$ is the outcome variable; the animal type (*control/HF*) a fixed effect; and the channel number ($1 - 16$) and the *animal ID* random effects. The analysis of $Entropy_{STD}$ follows by replacing 'MEAN' with 'STD'.

In *Equation 2*, the model $Entropy_{MEAN,EVENT}$ is shown and refers to entropy mean data within event regions where the model for mean entropy data outside event regions is $Entropy_{MEAN,NON-EVENT}$. In this way, models are constructed for event/non-event, mean/std entropy as the outcome variable; the event type (event/non-event), the animal type (control/HF), and coactivity computation type (mean/std) are fixed effects; and channel number, animal ID, and entropy (type matching the outcome entropy's type, mean, or std) are random effects.

For all analyses using mixed effects modeling, the $\beta$ coefficients (fixed effects estimates), *p*-values, effect sizes ($d_{RM}$ based on repeated measures Cohen's $d_{RM}$; *Lakens, 2013*), 95% CIs of $\beta$ coefficients (lower, upper bounds) are reported in results in ($\beta$, $\pm CI$, $d_{RM}$, $p$) format. The $\beta$ coefficients indicate the adjusted differences (units matching the outcome variable's unit) in one group compared to the other. For analyses with independent samples, *p*-values and independent samples effect sizes ($d$, based on Cohen's $d$) are reported in ($p$, $d$) format. For all analyses, a two-sided $p < 0.05$ denoted statistical significance.

# Acknowledgements

This work was funded by the National Institutes of Health, Office of The Director DP2 OD024323-01 and NHLBI R01 HL159001. NZG was funded by the National Science Foundation American Society of Engineering Education's Engineering Fellows Postdoctoral Fellowship Award ID #2127509. The authors would like to thank Prof. Jeffrey Ardell for fruitful discussions.

# Additional information

## Funding

| Funder | Grant reference number | Author |
| --- | --- | --- |
| National Institutes of Health | DP2 OD024323-01 | Olujimi A Ajijola |
| National Heart, Lung, and Blood Institute | R01 HL159001 | Olujimi A Ajijola |

| Funder | Grant reference number | Author |
|---|---|---|
| National Science Foundation | ASEE 2127509 | Nil Z Gurel |

The funders had no role in study design, data collection and interpretation, or the decision to submit the work for publication.

## Author contributions
Nil Z Gurel, Koustubh B Sudarshan, Conceptualization, Software, Formal analysis, Validation, Investigation, Visualization, Methodology, Writing – original draft, Writing – review and editing; Joseph Hadaya, Validation, Writing – review and editing; Alex Karavos, Software, Writing – review and editing; Taro Temma, Yuichi Hori, Data curation, Writing – review and editing; J Andrew Armour, Guy Kember, Resources, Supervision, Funding acquisition, Validation, Project administration, Writing – review and editing; Olujimi A Ajijola, Resources, Supervision, Funding acquisition, Validation, Investigation, Project administration, Writing – review and editing

## Author ORCIDs
Nil Z Gurel ⬤ http://orcid.org/0000-0002-3702-0449
Koustubh B Sudarshan ⬤ http://orcid.org/0000-0002-1256-1169
Olujimi A Ajijola ⬤ http://orcid.org/0000-0001-6197-7593

## Ethics
The study was performed under a protocol approved by the University of California Los Angeles (UCLA) Animal Research Committee (ARC), in compliance with the UCLA Institutional Animal Care and Use Committee (IACUC) guidelines and the National Institutes of Health (NIH) Guide for the Care and Use of Laboratory Animals (Protocol #: ARC 2015-022). For SG neural data collection, the animals were sedated with tiletamine and zolazepam (Telazol, 4-8mg/kg) intramuscularly, intubated, and maintained under general anesthesia with inhaled isoflurane (2%). Continuous intravenous saline (8h) was infused throughout the protocol and animals were temperature maintained using heated water blankets (37).At the end of the protocol, animals were euthanized under deep sedation of isoflurane and cardiac fibrillation was induced.

## Decision letter and Author response
Decision letter https://doi.org/10.7554/eLife.78520.sa1
Author response https://doi.org/10.7554/eLife.78520.sa2

---

# Additional files

## Supplementary files
• MDAR checklist

## Data availability
Data is available in the Dryad repository. Codes are at GitHub (https://github.com/Koustubh2111/Cofluctuation-and-Entropy-Code-Data; copy archived at swh:1:rev:1ca5e9ce38151715cfa1aeb3d38f3bdbbf796a05).

The following dataset was generated:

| Author(s) | Year | Dataset title | Dataset URL | Database and Identifier |
|---|---|---|---|---|
| Gurel NZ, Sudarshan K, Hadaya J, Karavos A, Temma T, Hori Y, Armour J, Kember G, Ajijola O | 2022 | Metrics of High Cofluctuation and Entropy to Describe Control of Cardiac Function in the Stellate Ganglion: Neural Recordings from Swine Models | https://doi.org/10.5068/D10Q22 | Dryad Digital Repository, 10.5068/D10Q22 |

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

## Appendix 1

### Cofluctuation and event rate definitions

#### Coactivity matrix

A 16×16 correlation matrix, 4×4 version is shown in *Figure 8B* for $n = 4$ channels, is used to investigate spatial coherence among neural populations in different regions of the SG spanned by 16 electrodes (*Appendix 1—figure 1*). The coactivity matrix at each timestamp is found from Pearson's cross-correlation between all possible pairs of spike rate, causal channel, sliding mean, and standard deviation. The sliding mean and standard deviation of spike rate are $Spike_{RateMEAN}$ and $Spike_{RateSTD}$, and are on the *y*-axis of *Figure 8A*. These are referred to as 'spike rate' in what follows when both are implied. To fix ideas, consider Pearson's cross-correlation coefficient ($R$) between channels 1 and 2, labeled as $R_{12}$: namely, the red and blue windows, respectively, in *Figure 8A*. In the coactivity matrix depicted in *Figure 8B*, there are $n = 4$ channels, hence $n - 1 = 3$ super-diagonals. These are vertically stacked in *Figure 8C* at each timestamp beginning with the first super-diagonal as $R_{12}$, $R_{23}$, and $R_{34}$. In this way, adjacent channels are placed at the bottom followed by super-diagonals corresponding to two and three channels of separation. The super-diagonal of the 16-channel LMA electrode correlation matrix has $n = 16$ channels separated by $500 \mu m$ and $n(n - 1)/2 = 120$ possible pairwise correlations (see *Appendix 1—figure 5* for an example). This yields 120 rows in the stacked version of the coactivity matrix at each timestamp analogous to the same visualized in *Figure 8C* for $n = 4$ channels.

#### Cofluctuations and event rate

The univariate cofluctuation time series is the percentage of coactivity matrix members, at each timestamp, that exceed a threshold Pearson's $R > C$, depicted in *Figure 8D*. Discrete events are considered to begin at a time of up-crossing of the univariate cofluctuation time series through a threshold $T$. Each event ends at a down-crossing some time later, as shown in *Figure 8E*. These discrete events capture spatiotemporal zones of high SG coactivity. Up-crossing times are, respectively, converted to an event rate ($ER_{MEAN}, ER_{STD}$) for the ($Spike_{RateMEAN}$, $Spike_{RateSTD}$) over a duration

$$(ER_{MEAN}, ER_{STD}) = \frac{(N_{MEAN}, N_{STD})}{EventsDuration} \tag{A1}$$

where event rate, $ER$, has units $1/s$ and ($N_{MEAN}, N_{STD}$) are the number of up-crossings within the *EventsDuration* considered.

#### Cofluctuation probability distribution

The cofluctuation time series at each threshold $C$ (as in *Figure 8D*) qualitatively approximates a log-normal distribution. The log-normal fits of cofluctuation time series (*Figure 2*) are obtained using Python SciPy package, with statistics and random numbers module (scipy.stats) (*Virtanen et al., 2020*).

#### Bootstrapping and selection of convergent thresholds

The event rate is calculated based on a pair of thresholds ($C, T$). The first threshold ($C$, *Figure 8D*) is used to reduce the coactivity time series of matrices to a univariate cofluctuation time series. The univariate series is the percentage of coactivity matrix entries exceeding $C$ at each timestamp. The cofluctuation time series is then used to define regions of high cofluctuation based on intervals where the time series exceeds a second threshold $T$. These regions are discrete 'events' that begin and end when the cofluctuation time series, respectively, up- and down-crosses through $T$ (*Figure 8D*). Bootstrapping of the event up-crossing timestamps is used to construct the event rate histogram of a threshold pair ($C, T$).

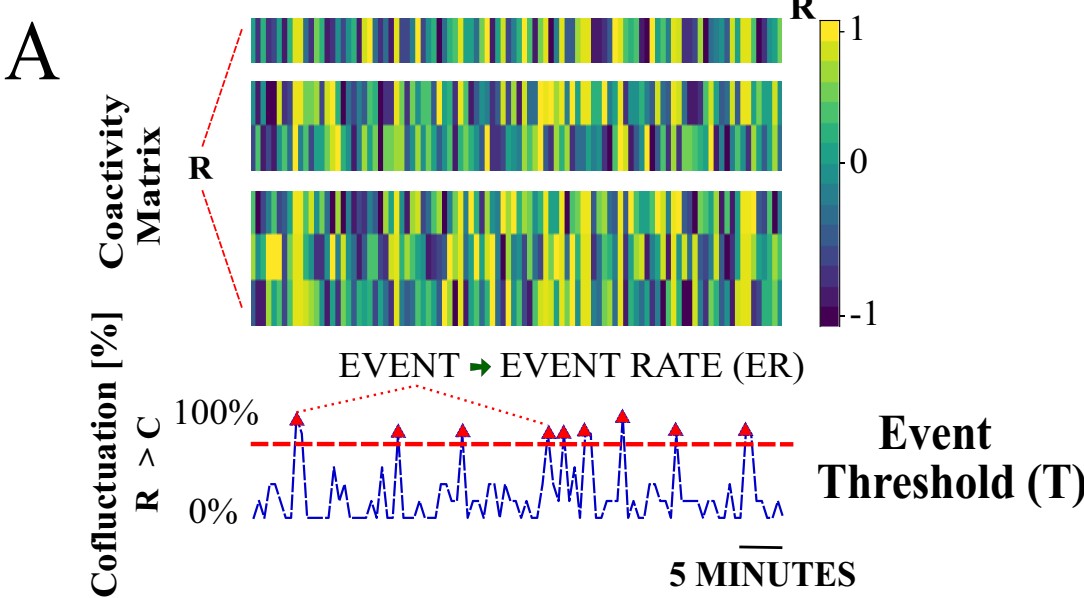

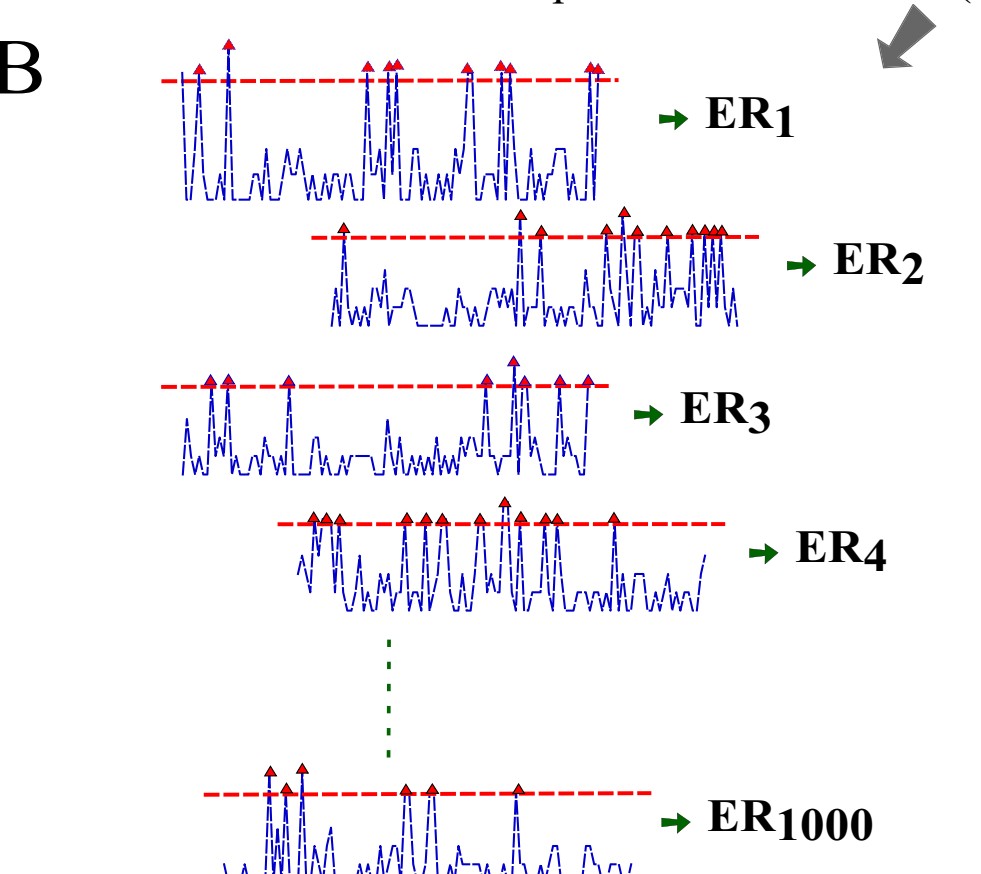

**Appendix 1—figure 1.** Cofluctuation bootstrapping pipeline for individualized event rate (ER) for each animal—Part I. (**A**) Coactivity matrix and cofluctuation time series for a cofluctuation threshold and event threshold pair $(C, T)$. (**B**) Cofluctuation time series with depicted events (red triangles are up-crossing timestamps) for a range of $(C, T)$ pairs. Panel A is further explained in **Appendix 1—figure 5**.

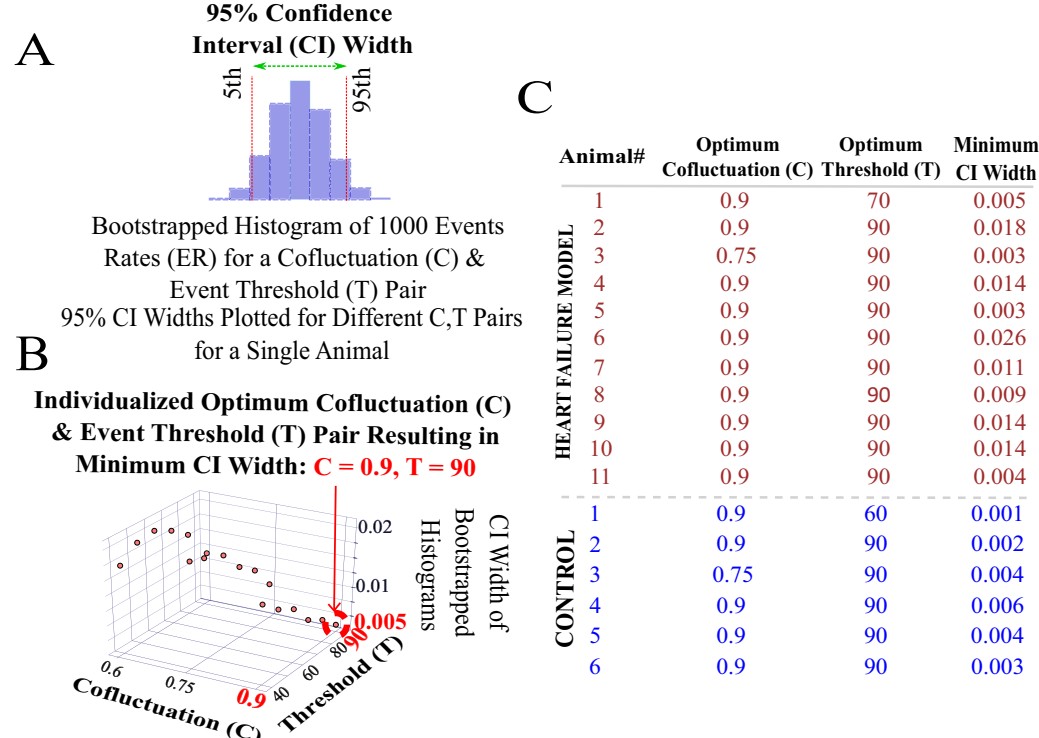

**Appendix 1—figure 2.** Cofluctuation bootstrapping pipeline for individualized event rate (ER) for each animal—Part II. (**A**) Bootstrapped histogram of ERs for a single $(C, T)$ pair with 95% confidence interval ($CI$) width. (**B**) Threshold pair $(C, T)$ is chosen for an animal given non-zero $ER$ and $ER$ convergence. Depicted $CI$ widths are found at all of the depicted 18 pairs available on the axes grid. (**C**) List of $(C, T)$ pairs that show $ER$ convergence with $CI$ widths for each animal.

These histograms lead to a convergent choice of threshold pairs $(C, T)$. The convergent $(C, T)$ pair is taken as the location in $(C, T)$ space where the CI width shows apparent convergence. An upper bound on $(C, T)$ is imposed so that there is sufficient data to compute the desired statistics.

The procedure is visualized in **Appendix 1—figure 1A** using a surrogate coactivity matrix $R$. Univariate cofluctuation time series are created from a range of thresholds $C$ that inclusively vary over 60–90% with 15% increments. Discrete events are determined, shown as red up-crossing triangles in **Appendix 1—figure 1A–B**, for each of the thresholds $C$ and considered over an inclusive range 40–90% with 10% increments of event thresholds $T$. Bootstrapped events provided the associated $ER$ histogram of each $(C, T)$ threshold pair and desired 95% CI width of each animal (**Appendix 1—figure 2**). A convergent $(C, T)$ pair for an animal is provided in **Appendix 1—figure 2B** $(C, T = 0.9, 90)$, that converged to a 95% CI width of 0.005. Following this approach, convergent $(C, T)$ pairs and bootstrapped CI widths are listed for each animal in **Appendix 1—figure 2C**.

Using these individualized convergent $(C, T)$ pairs, original (i.e., not bootstrapped) data are used to calculate event rates for each animal. Note that event rates are calculated from both spike rate mean and standard deviation coactivity matrices, and referred to as $ER_{MEAN}$ and $ER_{STD}$. These are then used in statistical analyses (one $ER_{MEAN}$ and one $ER_{STD}$ per animal) as shown in **Figure 7**. A similar procedure was performed in the literature using neuroimaging time series data based on Pearson's $R$ (**Zamani Esfahlani et al., 2020**); however, the threshold selection process was qualitative. In this work, we have developed a quantitative approach for threshold selection.

## Neural specificity

The neural specificity metric (**Sudarshan et al., 2021**), **Appendix 1—figures 3 and 4**, is used to evaluate the degree to which neural activity is biased toward control target states taken here as LVP. Briefly, this metric is computed in three stages

1. *Neural sampling*: The value of the target state (LVP) is 'sampled' at the timestamp of each spike occurrence. This sampling is assumed to approximate a quasi-stationary distribution over a causal (backward in time) sliding window of spiking activity that is updated at each new timestamp. The distribution is approximated as a normalized and sliding histogram of neurally sampled target states (LVP).

2. *Random sampling*: The normalized, sliding random sampling histogram is found at each spike occurrence in (1), but based on *all* available LVP samples within the same causal window referenced in (1), which approximates the random sampling limit.

3. *Neural specificity*: The normalized, sliding random sampling histogram (2) is subtracted from its neural sampling counterpart (1) to form the neural specificity contrastive measure.

Subtraction of the random sampling histogram from the neural sampling histogram allows for the discovery of the degree to which neural activity is biased, or specific, toward sampling control target states (LVP here) relative to random sampling. To explain the construction of the metric with LVP, a representative window is shown in *Appendix 1—figure 3A* with the spikes shown as green dots over LVP waveform. The following steps outline the construction of the neural specificity metric, $A$, for a representative LVP window

1. *Neural sampling*: Following (*Sudarshan et al., 2021*), the normalized sliding window histogram of neurally sampled $LVP_j$ at all spike times tj and taken over $M$ bins is defined for bin $k$ as

$$H(SLVP_j)_k = \frac{h(SLVP_j)_k}{\Sigma_{k=1}^{k=m} h(SLVP_j)_k} \tag{A2}$$

   (*Equation A2*) approximates the distribution of neural sampling of the target LVP at the green dots over a causal window in *Appendix 1—figure 3A*. The resulting normalized histogram shown for one timestamp (green line) in *Appendix 1—figure 3*.

2. *Random sampling*: The normalized sliding window histogram at the random sampling limit of $LVP_j$ is computed as in (1), but based on *all* LVP samples within the same causal window and defined as $H(LVP_j)_k$. This is depicted as sampling of the pink line in *Appendix 1—figure 3A* over the same causal window used to describe neural sampling of LVP. The result is shown for one timestamp as the normalized histogram (pink line) in *Appendix 1—figure 3B*.

3. *Neural specificity*: The neural specificity, $A_{jk}$, for bin $k$ is

$$A_{jk} = H(SLVP_j)_k - H(LVP_j)_k \tag{A3}$$

   $A_{jk}$ is mapped to three levels (*less*, *same*, *greater*) relative to random sampling. These are respectively defined as $(-1, 0, 1)$ and depicted as (*purple*, *teal*, *yellow*) in *Appendix 1— figures 3C and 4AFigure 4*. As such, given the mapping threshold $\alpha > 0$ it follows that $(A_{jk} < -\alpha, A_{jk} < \alpha, A_{jk} > \alpha)$ is, respectively, $(-1, 0, 1)$ implying (*less*, *same*, *greater*) neural specificity relative to random sampling and visually represented as (*purple*, *teal*, *yellow*).

## Entropy definitions

### Entropy

The neural specificity is reduced from a multivariate signal to a univariate signal by computing the Shannon entropy at each timestamp of the mapped neural specificity metric (*Figure 7*, *Equation A3* mapping description). The entropy of the absolute change between adjacent normalized histogram bins is a measure of coherence in neural specificity. The absolute change in the mapped $A_{jk}$ at time $t_j$ and between adjacent bins $(k, k+1)$, $k = 1, ..., m-1$ is the set $\Delta A_j = (0, 1, 2)$ with members $\Delta A_{ji}, i = 1, 2, 3$. Using a base 3 logarithm to scale the entropy between 0 and 1, the entropy $E_j$ of the difference in the mapped $A_{jk}$ at each timestamp $t_j$.

$$E_j = -\Sigma_{\Delta A_{ji}=1}^3 p(\Delta A_{ji}) \ln_3(p(\Delta A_{ji})) \tag{A4}$$

This unequally sampled series is interpolated to the equally sampled time series $E$.

### Event entropy

The neural specificity is a measure of specificity, or bias, of neural activity to target states. However, unusually high and short-lived cofluctuations indicate intervals in time, or 'events', when coactivity

between channel pairs implies that SG processing has undergone sudden changes. Functional relevance of cofluctuation events is found by considering the extent to which neural specificity to the target (LVP here) is similar or different inside and outside these events.

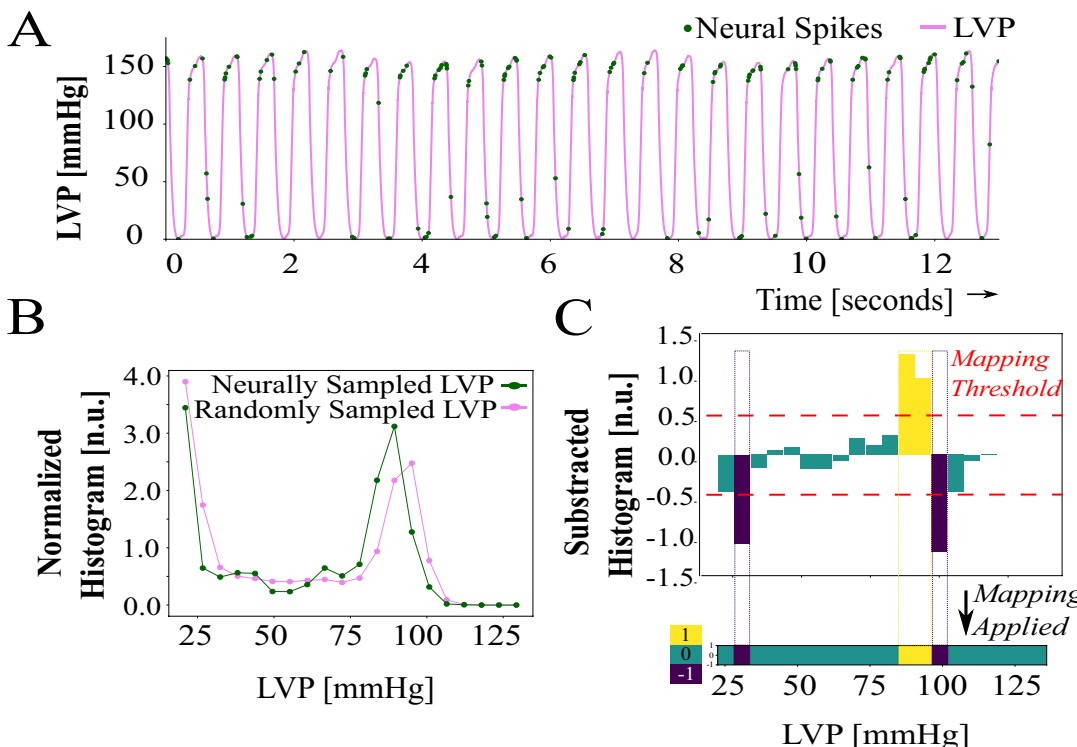

**Appendix 1—figure 3.** Neural specificity and entropy computation—Part I. (**A**) Neural specificity sample showing left ventricular pressure (LVP) and neural spikes. Spiking activity is more specific or biased (yellow), over random sampling, to LVP just below systolic pressures. (**B**) Normalized histograms of random and neurally sampled LVP. (**C**) Bars show subtracted histograms and colors indicate the specificity thresholded with $\alpha = 0.5$: specificity exceeding $\alpha$ is yellow, below $-\alpha$ is blue, and between the bounds $(-\alpha, \alpha)$ is teal.

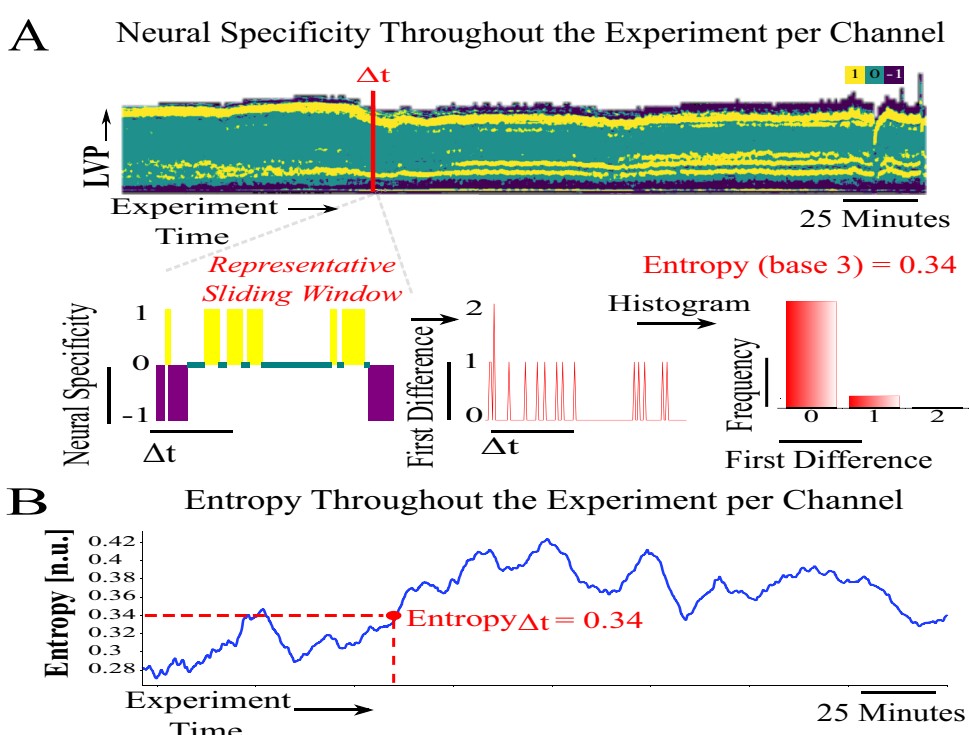

**Appendix 1—figure 4.** Neural specificity and entropy computation—Part II. (**A**) Sliding neural specificity time series (top) for a selected sliding window width $\Delta t$. Entropy of neural specificity, computed from red highlighted window of width, $\Delta t$ absolute difference (shown for a sample at bottom). (**B**) Entropy time series corresponding to the overall experiment, entropy sample computed from steps in (**D**). An animated version is in Appendix 1—Animation 1.

# Coactivity Matrix from Sliding Mean Spike Rate

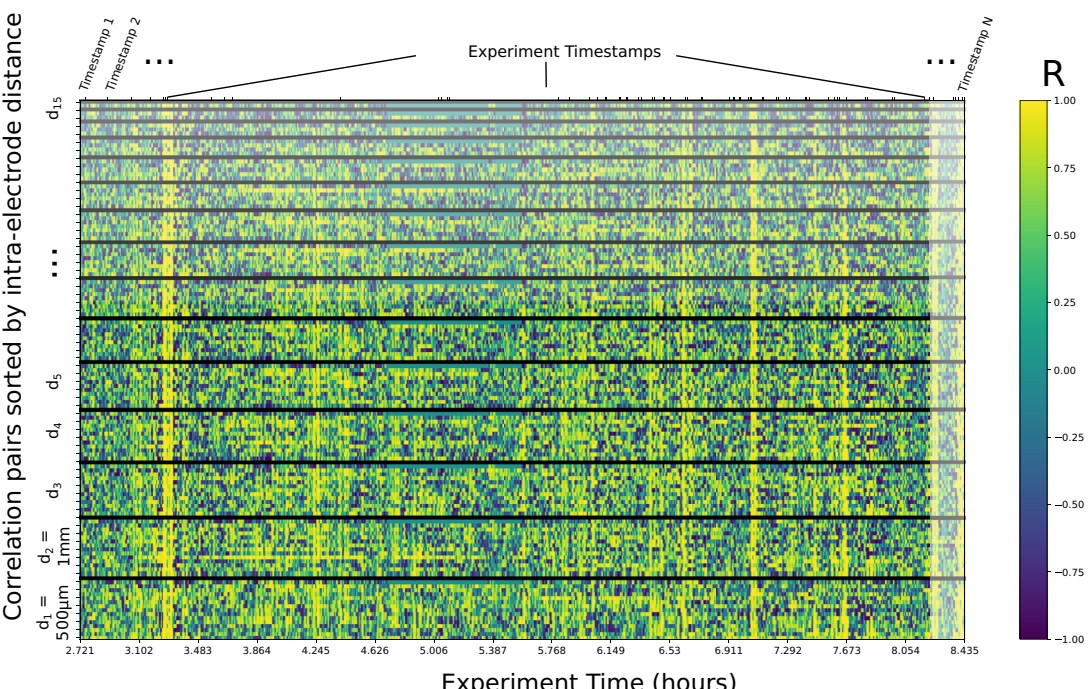

**Appendix 1—figure 5.** 16-Channel version of *Appendix 1—figure 1A*. A sample coactivity matrix computed from the sliding (rolling) mean of spike rate for the 16-channel linear mapping array (LMA) used throughout an experiment. *Y* axis shows correlation pairs (i.e., Channel 1 vs. Channel 2). Correlation pairs are stacked such that the lowest row corresponds to channels separated by the minimum inter-electrode distance ($d_1 = 500\mu m$, 1-electrode away), and inter-electrode distance increasing from a separation of 1–15 channels at the top. For instance, the highest row shows the pair separated by the maximum inter-electrode distance ($d_{15} = 15 * 500\mu m = 7.5mm$, 15-electrodes away). The order corresponds to the super-diagonals of the 16×16 correlation matrix. *Y* axis includes 120 comparisons for 16 channels, colors indicate Pearson's correlation coefficients specified in the color legend.

Therefore, the functional relevance of cofluctuations in SG neural activity is examined by breaking the time-evolution of entropy of neural specificity into regions: 'event' regions (within event intervals) and 'non-event' regions (outside event intervals). The mean and standard deviation of event and non-event entropy time series per channel are computed for each experiment and collectively referred to as 'event entropy' where this is convenient.

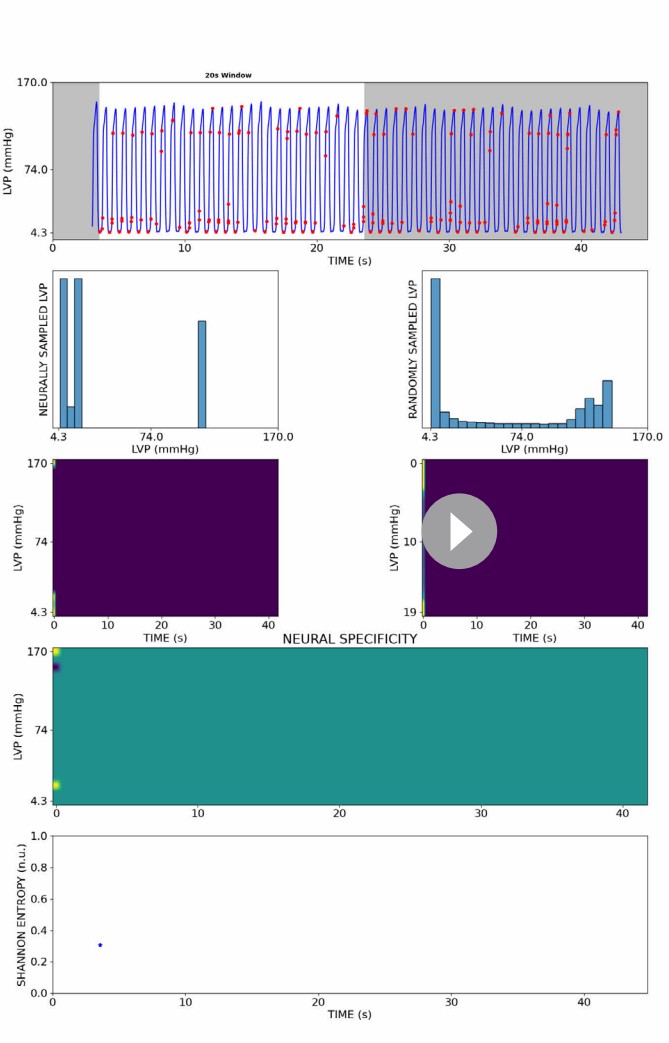

**Appendix 1—animation 1.** Entropy animation. The Animation 1.gif file contains an animation of the building of the neural specificity metric with respect to left ventricular pressure (LVP). Row 1: Computation of the neural specificity metric for different frames is shown in the form of 20 s moving windows on a 45-s segment of the LVP data. The LVP data are shown in blue tracings with the neural spikes represented as red dots. Row 2: Two normalized histograms for each of the moving windows are calculated and shown in the second row. The histogram of the LVP computed at spike times (neurally sample LVP) is on the left. The histogram of the LVP in the window (randomly sampled LVP) is on the right. Row 3: The computed histograms are then used to compute two matrices in the third row. The matrices contain all the corresponding histograms computed in the previous step arranged vertically with a hard threshold of 0.5 applied, that is, histogram values greater than 0.5 are set to 0.5 (colored yellow). Row 4: The two matrices computed in row 3 are subtracted to obtain the neural specificity metric shown in the fourth row. The color scheme is explained in Appendix 1—figure 4. Row 5: Entropy shown in the fifth row is obtained by calculating the Shannon entropy (such as depicted in Appendix 1—figure 4) for the subtracted histogram in each of the moving windows for the duration of the metric.

