## [Editor Report]

The paper thoroughly examines the role of the stellate ganglia in the control of cardiac rhythmicity. The empiric findings on stellate-ganglia-mediated regulation of cardiac activity in the presence of chronic cardiac failure, in particular, are translationally relevant and could be the basis of future drug-based interventions.

---

## [Decision Letter]

**Decision letter after peer review:**

Thank you for submitting your article "Metrics of High Cofluctuation and Entropy to Describe Control of Cardiac Function in the Stellate Ganglion" for consideration by *eLife*. Your article has been reviewed by two peer reviewers, one of whom is a member of our Board of Reviewing Editors, and the evaluation has been overseen by a Senior Editor. The reviewers have opted to remain anonymous.

The reviewers have discussed their reviews with one another, and the Reviewing Editor has drafted this letter to help you prepare a revised submission.

Essential revisions:

1. Please rewrite the Abstract and Conclusion section to convey what a 'real world' study outcomes mean to cardiac function, and its relevance to potential cardiac therapy in the future. Currently, the conclusions are broad and vague.

2. Please reduce jargon to make the manuscript more readable to a general audience. E.G. "event entropy" is not easy to interpret physiologically and is just one of many examples.

3. Confirmation that resiniferatoxin killed the afferents and discussion of difference to data previously reported by Zucker's group, for example.

4. Indicate whether the single unitary activity was discriminated prior to analysis or whether multiple spikes analysed as events and what proportion of events were cardiac-related versus non-cardiac given the multiple targets of the stellate, and show some non-cardiac-related data as a comparator; was any unitary activity related to respiration, for example? Please also confirm that the cardiac-related data was not a mechanical-movement artefact and how this possibility was excluded.

5. Caveats should be discussed including effects of general anaesthesia, open chest, and open pericardial sac effects on neural activity caused by potential changes in pulmonary physiology.

6. Make sure all Figures are labelled and have axes and that the legends are written to assist a non-specialist's understanding; clarity could be improved in some by expanding compressed regions.

7. Self-citation is high and we recommend broadening the literature cited.

*Reviewer #1 (Recommendations for the authors):*

1. Could the authors indicate whether the single unitary activity was discriminated prior to analysis or whether multiple spikes analysed as events.

2. It would be helpful to show some data which showed no relationship with left ventricular pressure as a comparator.

3. What was used to hold and position the single shank microelectrode array into the ganglion? Was this left free floating after positioning in the ganglia? If not, how can the authors be sure that the activity related to left ventricular pressure is not generated by the movement artefact of the heart/arterial pulsing causing mechanical stimulation of the neurons? Additional description is needed.

4. The authors state they measured respiratory pressure. Given that the respiratory system couples to the sympathetic nervous system and is a major contributor to sympathetic activity generation in heart failure, why did the authors not also look at the neural specificity relevant to the respiratory cycle?

*Reviewer #2 (Recommendations for the authors):*

Completely rewrite the Abstract to convey the results and the conclusions of the work and what they mean to cardiac function.

It is inappropriate to report differences in mean "event rates" without stating those means and instead reporting on the p-values. L417-18.

The respiration/ventilator cycle produces one of the most consistent cyclic changes in sympathetic activity including the cited publication (Sudarshan et al., 2021). Why is this not assessed here?

---

## [Author Response]

Essential revisions:1. Please rewrite the Abstract and Conclusion section to convey what a 'real world' study outcomes mean to cardiac function, and its relevance to potential cardiac therapy in the future. Currently, the conclusions are broad and vague.

We thank the reviewers for this helpful suggestion. We revised the abstract and conclusions. In the revised abstract, we addressed the current limitations of therapies targeting stellate ganglia (SG) and how understanding of information processing in SG may advance the current therapies. In the revised conclusions, we restated how the metrics developed in the manuscript can be used to assess the dynamic information processing in SG. We cited a recent manuscript that extensively reviews how experiments and high-density recording technologies may be instrumental in studying these information dynamics, compared to traditional studies in this area spanning from 1980s to 2000s. Until 2000s, this field was dominated by single-unit recordings with analysis of simple metrics such as firing rate and stimulation latency. Linkages between neural recordings (inputs) and peripheral outputs (blood pressure) have never been studied or systematically characterized.

Revised Abstract

“Stellate ganglia within the intrathoracic cardiac control system receive and integrate central, peripheral, and cardiopulmonary information to produce postganglionic cardiac sympathetic inputs. Pathological anatomical and structural remodeling occurs within the neurons of the stellate ganglion in the setting of heart failure. A large proportion of SG neurons function as interneurons whose networking capabilities are largely unknown. Current therapies are limited to targeting sympathetic activity at the cardiac level or surgical interventions such as stellectomy, to treat heart failure. Future therapies that target the stellate ganglion will require understanding of their networking capabilities to modify any pathological remodeling. We observe SG networking by examining cofluctuation and specificity of SG networked activity to cardiac cycle phases. We investigate network processing of cardiopulmonary transduction by SG neuronal populations in porcine with chronic pacing-induced heart failure and control subjects during extended in-vivo extracellular microelectrode recordings. We find that information processing and cardiac control in chronic heart failure by the SG, relative to controls, exhibits: (i) more frequent, short-lived, high magnitude cofluctuations, (ii) greater variation in neural specificity to cardiac cycles, and (iii) neural network activity and cardiac control linkage that depends on disease state and cofluctuation magnitude.” (Abstract, Page 1)

Revised Conclusions

“In this study, we looked, for the first time to our knowledge, at long-term studies of in vivo cardiac control in baseline states. The baseline states provide unique signatures that differentiate animals with heart failure and controls. We discovered the inputs (i.e., neural signals) and outputs (i.e., blood pressure) are linked, which led us to develop metrics to analyze the dynamical state of this networked control [Gurel et al., 2022]. The primary observation has been that event-based processing within the stellate ganglion and its relationship to cardiac control is strongly modified by heart failure pathology. Our analysis is pointing to heart failure being best considered as a spectrum rather than a binary state. The magnitude of cofluctuation and neural specificity may give us a measure of the degree of heart failure and insight the extent to which cardiac control is compromised with respect to neural specificity and/or cofluctuation. Future therapies may benefit from being able to infer the degree of heart failure in terms neural markers as represented in this work, in a less invasive way.

Intriguing connections involve the alignment of our work with a growing consensus in neuroscience. Changes in neural activity through space and time are associated with the onset and development of specific pathologies. For instance, spatiotemporal brain-wide cofluctuations were reported to reveal major depression vulnerability [Hultman et al., 2018]. Another study reported that brain’s functional connectivity is driven by high-amplitude cofluctuations and that these cofluctuations encode subject-specific information during experimental tasks [Esfahlani et al., 2020]. These parallel conclusions in cardiac and neuroscience space point that similar experimental methods will be required to measure neural control integration. These measurements will be instrumental to design and assess the efficacy of neurally-based clinical interventions both at the level of the brain and the stellate ganglion.” (Conclusion, Pages 8-9)

2. Please reduce jargon to make the manuscript more readable to a general audience. E.G. "event entropy" is not easy to interpret physiologically and is just one of many examples.

For focusing on experimental methods and implications of the results, we have moved all mathematical explanations of the used metrics to appendix section, including sections “Cofluctuation and Event Rate Definitions”, “Neural Specificity”, and “Entropy Definitions” (Pages 5-9 in old manuscript). After outlining experimental details and signal processing pipeline, we detailed how these metrics have been evaluated with statistical methods between animals with HF and controls. We referred to appendix information when necessary and simplified the information text. The reviewers can see the Appendix 1 (Pages 18-25 in the updated manuscript). Note that we also moved the methods figures under Appendix 1 (Figures 1, 2, 3, 4, 5).

3. Confirmation that resiniferatoxin killed the afferents and discussion of difference to data previously reported by Zucker's group, for example.

We thank the reviewers for reminding this clarification. The HF model animals were from our group’s experiments cited as “Cardiac afferent signaling partially underlies premature ventricular contraction–induced cardiomyopathy”, authored by Hori et al. The aim of this study was to determine whether cardiac transient receptor potential vanilloid-1 (TRPV1) afferent signaling promote premature ventricular contraction (PVC)–induced cardiomyopathy (PIC). PIC swine models were created via implanted pacemaker, and cardiac TRPV1 afferent fibers were selectively depleted using percutaneous epicardial application of RTX.

Depletion of cardiac TRPV1 afferent fibers by RTX was analyzed both structurally and functionally. To confirm the structural depletion of cardiac TRPV1 afferent, authors performed immunohistochemistry studies of the left ventricle (LV) and T1 dorsal root ganglion (DRG). Calcitonin gene-related peptide (CGRP)-immunoreactive fibers, a marker of sensory afferent nerves, was significantly reduced within the nerve bundles located in the LV for the RTX-treated group at both 4 weeks and 8 weeks (Author response image 1). Furthermore, the depletion of cardiac TRPV1 afferents was confirmed by the significant reduction of CGRP-expressing neurons in DRG (Author response image 1). To confirm functional depletion, the response to the agonist of TRPV1 channel bradykinin and capsaicin was examined. The RTX-treated group had a significantly lower LV pressure (LVP) response in the application of bradykinin and capsaicin in 4 weeks (Author response image 1, P = 0.028 and P = .028, respectively) and 8 weeks (Author response image 1, P = .041 and P = .015, respectively), indicating that elimination of cardiac sympathetic afferent reflex was accomplished by RTX application in each case.

**Author response image 1. sa2fig1:** Structural and functional TRPV1 afferent depletion by RTX, from Hori et al. (**A**): Representative images of LV nerve fibers demonstrating depletion of cardiac TRPV1 channel. Depletion of cardiac TRPV1 channel was confirmed by expression of CGRP. PVC 8W group (top row) shows double immunostaining for PGP9.5 and CGRP in LV nerve fibers (solid arrows), whereas the RTX-treated group (bottom row) showed no CGRP expression in nerve fibers (arrows). (**B**): Percentage of CGRP-expressing nerve fibers in the total of nerves confirmed in each group. The RTX-treated group showed a significant decrease of CGRP expression in both the 4-week and 8-week PIC models. (**C**): Representative images of decreasing CGRP-expressing neurons in DRG T1 due to cardiac TRPV1 channel depletion. (**D**): Percentage of CGRP-expressing neurons in DRG. CGRP expression significantly decreased with RTX treatment at both 4 weeks and 8 weeks. (**E**): Functional TRPV1 afferent depletion was confirmed by the response to the agonist of TRPV1 channel bradykinin and capsaicin. The RTX-treated group showed significantly weaker hemodynamic changes compared to the nontreated group for both of the agonists of the TRPV1 channel, and in 4-week and 8-week data. *P<0.05; **P<.01. Abbreviations: 4W = 4 weeks; 8W = 8 weeks; CGRP = calcitonin gene-related peptide; DRG = dorsal root ganglia; LV = left ventricle; LVP = left ventricular pressure; PGP9.5 = protein gene product 9.5; TRPV = transient receptor potential vanilloid-1.

We added this information under Methods – Animal Experiments section and cited the reference in the updated manuscript.

“We confirmed the RTX depleted the afferents by analyzing both structural and functional data [Hori et al., 2021]. Structural depletion was proven with immunohistochemistry studies of the left ventricle (LV) and T1 dorsal root ganglion (DRG). Calcitonin gene-related peptide (CGRP)immunoreactive fibers, a marker of sensory afferent nerves, was significantly reduced within the nerve bundles located in the LV for the RTX-treated group. Furthermore, the depletion of cardiac transient receptor potential vanilloid-1(TRPV1) afferents was confirmed by the significant reduction of CGRP-expressing neurons in DRG. Functional depletion was proven by the response to the agonist of TRPV1 channel bradykinin and capsaicin. The RTX-treated group had a significantly lower LV pressure (LVP) response in the application of bradykinin and capsaicin, indicating that elimination of cardiac sympathetic afferent reflex was accomplished by RTX application in each case.” (Methods – Animal Experiments, Page 11)

4. Indicate whether the single unitary activity was discriminated prior to analysis or whether multiple spikes analysed as events…

We thank Reviewer #1 for their time and comments. Our experiments can be classified as “multi-unit recordings”. For single-unit recordings, as the reviewer suggested, the target neurons must be isolated and recording electrodes should be fine-tipped with low-impedance conductors for high quality recording. Single-unit recordings may record several isolated neurons with wire electrodes in separate nerve bundles (Boczek-Funcke et al., 1992). While large electrode arrays increase the amount of collected information per unit time, they may not provide sufficient isolation. The multi-unit signals, such as the 16-channel linear microelectrode array recordings in our paper, involve recording of closest neural populations, rather than the closest single neuron. In the recent neuroscience literature, a shift in the experimental focus to interactions of neural populations and their ensemble behaviors (Yuste, 2015; Zamani Esfahlani et al., 2020) has led to the nearly exclusive use of multi-unit recordings. A review of single vs. multi-unit literature can be seen in Table 1 of Gurel et al., there is a shift in recording neural populations with multi-unit electrodes in recent years in this area.

We added this important experimental detail to the manuscript under Methods-Animal Experiments:

“It should be noted that our study deals with multi-electrode recordings of the closest neural populations to the electrode array. The earliest fundamental studies probing into cardiac nervous system used single-unit recordings, for which the target neurons should be isolated and appropriate low-impedance conductors should be used for obtaining high quality neural signals. Unlike these early studies, we used multi-unit (16-channel) electrode arrays to monitor the ensemble behaviors of SG neural populations. This experimental shift from single-unit to multiunit recording has gained interest in the recent years in neurocardiology and neuroscience communities, offering an experimental view to the ensemble behaviors of neural populations [Gurel et al.]” (Methods – SG Neural Recordings and Experimental Protocol, Page 11)

…what proportion of events were cardiac-related versus non-cardiac given the multiple targets of the stellate, and show some non-cardiac-related data as a comparator;

We thank the reviewer for raising this point. We would like to clarify that, in this study we compared the extent of network processing within SG neural populations in control and HF animals using novel metrics. The detected events/spikes were not classified based on their target as cardiac or non cardiac.

This point was also addressed in our previous study “A Novel Metric Linking Stellate Ganglion Neuronal Population Dynamics to Cardiopulmonary Physiology**”** by Sudarshan et al. This study looked at multiple channels of continuous recordings from the porcine stellate ganglion along with LVP and respiratory pressure (RP) (Figure R2A). SG activity across many channels showed an integration of cardio-pulmonary function. Channel activities were modulated by the respiratory cycle as seen by the increase in subsequent spike interval or spike period above a pressure of 0 cmH2O compared to at or near the same pressure (Respiratory pressures above 0cmH_2_O indicated by yellow blocks in Author response image 2, B and C and spike period shown as the heavy blue line in R2C). A subset of channels also contained spikes phase locked to near peak or peak LVP at or near respiratory pressures of 0cmH_2_0 (The spikes shown as red dots and LVP shown as blue tracings in Author response image 2). The cyclic nature of the spike period was also abolished during a brief period of apnea indicating a breathing influence on the spike period (Author response image 2). These findings show that stellate population dynamics are influenced by an integration of cardiopulmonary information. The extent of such integration and processing of cardiopulmonary information by SG population as a whole was explored in this study. Future experiments will be aimed at labeling the detecting spikes based on a target such as cardiac and respiratory. We hope this answers the reviewer’s question.

**Author response image 2. sa2fig2:** Cardiopulmonary integration is reflected in stellate ganglion neural activity [Sudarshan et al.2021]. (**A-B**): Representative recordings from several electrodes (blue and red tracings) using the 16channel linear array, along with respiration (Resp), left ventricular pressure (LVP), and the electrocardiogram (ECG) in purple tracings. Yellow bars highlight respiration (panel **A**) and the cardiac cycle (panel B). Activity in the blue and red channels are locked to cardiac and/or pulmonary. (**C**): Stellate ganglion (SG) neuron activity (red dots are individual spikes) shows increased firing rate at peak and near-peak LV pressures, however firing is inhibited during respiration, as reflected by increased spike period (heavy blue line) mirroring inspiration and expiration. (**D**): Representative response of SG neural activity to apnea (60 seconds). Black trace is scaled up respiratory activity, and black arrows identify baseline peak and trough of the spike period (green trace). (**E**): Spike period oscillation (peak – trough values) while breathing at baseline, and over the same period in apnea. n=6, *p=0.016, two-tailed Wilcoxon rank sum test.

Please also confirm that the cardiac-related data was not a mechanical-movement artefact and how this possibility was excluded.

This is an important question. In these experiments, the LMA cable was carefully placed in a manner where it was not susceptible to cardiac motion. Additionally, at the end of some experiments, the heart was in fibrillation, essentially immobilizing it. This actually led to an increase in firing frequency of stellate ganglion neurons.

5. Caveats should be discussed including effects of general anaesthesia, open chest, and open pericardial sac effects on neural activity caused by potential changes in pulmonary physiology.

We agree with the reviewer. We have added the following to limitations on page 14 of the manuscript.

“We cannot exclude possible effects of general anesthesia, open chest and open pericardial effects on our findings, though the effects are likely consistent across the groups studied in the same manner”. (Discussion, Page 8)

6. Make sure all Figures are labelled and have axes and that the legends are written to assist a non-specialist's understanding; clarity could be improved in some by expanding compressed regions.

Thank you for this suggestion. For improving clarity of some of the figures, we have split figures 4 and 5 in old manuscript into two parts in the updated manuscript. In addition, all figures are completely labeled with clear figure legends.

The changes are made in Appendix 1, pages 19-20 and 23-24:

Appendix 1, Figure 1: Cofluctuation bootstrapping pipeline for individualized event rate (ER) for each animal – Part I

Appendix 1, Figure 2: Cofluctuation bootstrapping pipeline for individualized event rate (ER) for each animal – Part II

Appendix 1, Figure 3: Neural specificity and entropy computation – Part I

Appendix 1, Figure 4: Neural specificity and entropy computation – Part II.

7. Self-citation is high and we recommend broadening the literature cited.

We thank the reviewer for pointing this out. We have added four additional citations that include methods such as neural population bias and spatiotemporal dynamics linkages to control targets in the neuroscience literature. We have added these citations to page 15 in the “Conclusion” section of the manuscript. In addition, it is our group’s specialty to carry these cardiac nervous system experiments, we are not aware of another group collecting multielectrode array data from the cardiac nervous system and studying population dynamics of cardiac neurons. Hence we build on based on our previous learnings. The most relevant literature (not necessarily related to cardiac nervous system) can be found in the neuroscience references we cited that contain applications of neural population recordings for different brain areas, mainly in neuropsychiatry domain to understand disease dynamics. (Conclusion, Page 9)